# Dual Box Embeddings for the Description Logic $\mathcal{EL}^{++}$

## ABSTRACT

OWL ontologies, whose formal semantics are rooted in Description Logic (DL), have been widely used for knowledge representation. Similar to Knowledge Graphs (KGs), ontologies are often incomplete, and maintaining and constructing them has proved challenging. While classical deductive reasoning algorithms use the precise formal semantics of an ontology to predict missing facts, recent years have witnessed growing interest in *inductive* reasoning techniques that can derive *probable* facts from an ontology. Similar to KGs, a promising approach is to learn ontology embeddings in a latent vector space, while additionally ensuring they adhere to the semantics of the underlying DL. While a variety of approaches have been proposed, current ontology embedding methods suffer from several shortcomings, especially that they all fail to faithfully model one-to-many, many-to-one, and many-to-many relations and role inclusion axioms. To address this problem and improve ontology completion performance, we propose a novel ontology embedding method named Box²EL for the DL $\mathcal{EL}^{++}$, which represents both concepts and roles as boxes (i.e., axis-aligned hyperrectangles), and models inter-concept relationships using a bumping mechanism. We theoretically prove the soundness of Box²EL and conduct an extensive experimental evaluation, achieving state-of-the-art results across a variety of datasets on the tasks of concept subsumption prediction, role assertion prediction, and approximating deductive reasoning. As part of our evaluation, we introduce a novel benchmark for subsumption prediction involving both named concepts and complex concepts defined with logical operators.[1]

## CCS CONCEPTS

• **Information systems** → *Web Ontology Language (OWL)*; • **Computing methodologies** → **Description logics**; **Ontology engineering**; **Learning latent representations**; *Statistical relational learning*.

## KEYWORDS

Ontology Embedding, Ontology Completion, Description Logic, Web Ontology Language, Link Prediction

**ACM Reference Format:**
Anonymous Author(s). 2018. Dual Box Embeddings for the Description Logic $\mathcal{EL}^{++}$. In *Proceedings of Make sure to enter the correct conference title from your rights confirmation emai (Conference acronym 'XX)*. ACM, New York, NY, USA, 13 pages. https://doi.org/XXXXXXX.XXXXXXX

---

[1]Code and data will be made available online.

---

*Conference acronym 'XX, June 03–05, 2018, Woodstock, NY*
© 2018 Association for Computing Machinery.
ACM ISBN 978-1-4503-XXXX-X/18/06...$15.00
https://doi.org/XXXXXXX.XXXXXXX

## 1 INTRODUCTION

Ontologies are a widely used formalism to represent general knowledge about a domain [37]. They are usually specified in the Web Ontology Language (OWL) [12] standard developed by W3C[2], and have been widely adopted in many domains such as the Semantic Web [15], healthcare [34], bioinformatics [16], and geoinformatics [46]. OWL allows for the expression of a variety of statements, ranging from simple relational facts to specifying concept hierarchies and complex logical relationships, and is underpinned by Description Logic (DL) [4] to define its formal semantics.

Although many real-world OWL ontologies have been developed and used with great success, such as the Gene Ontology GO [2] and the food ontology FoodOn [9], both maintaining these existing OWL ontologies and creating new ontologies has proved challenging and relies mostly on manual labor carried out by experts. Common ontology curation tasks include completing missing subsumptions between concepts (or membership relations between individuals and concepts) and identifying missing logical restrictions between concepts. Symbolic logical reasoning algorithms such as HermiT [11] and ELK [19] help address this problem by deductively inferring implicit knowledge from the precise semantics of an ontology, but this classical reasoning is often too rigid for real-world OWL ontologies and cannot derive knowledge that is only *probable* from the given data.

At the same time, there has been growing interest in representation learning-based methods for completing Knowledge Graphs (KGs) [17], i.e. relational facts in the form of RDF[3] triples <Subject, Predicate, Object>. Most of these approaches first learn structure-preserving *embeddings* of the entities and relations (predicates) of a KG in a latent vector space and then use them to score the likelihood of novel facts [18, 40]. For example, the classic method TransE [6] maps entities and relations to vectors such that translating the subject embedding by the relation embedding approximately yields the object embedding.

Similar embedding-based techniques as for KGs have been developed for inductive reasoning in ontologies, which promises to complement classical deductive reasoning for ontology curation tasks. Some approaches such as OPA2Vec [35] and OWL2Vec* [8] rely on exploiting textual meta information (e.g., concept labels and comments) to model similarities between entities, but do not retain the semantics defined by the underlying DL. Other approaches aim to directly embed the logical information of an OWL ontology in the latent space [22, 25, 31, 44], mostly targeting the OWL 2 EL profile [21], whose semantics are defined according to the DL $\mathcal{EL}^{++}$ [3]. Prominent examples include ELEm [22] and its extension EmEL++[25], which model concepts as high-dimensional balls, but fail to faithfully capture concept conjunction, since the intersection of two balls is no longer a ball. This led to the development of the state-of-the-art methods BoxEL [44] and ELBE [31], which

---

[2]https://www.w3.org/OWL/
[3]Resource Description Framework. https://www.w3.org/RDF/.

instead represent concepts as boxes (i.e., axis-aligned hyperrectangles). However, all of these approaches still rely on TransE [6] to model roles (i.e., binary relations) as simple translations, which is unable to capture one-to-many, many-to-one, or many-to-many relationships [23, 42], and is limited in its ability to faithfully represent inclusion relationships between roles. Furthermore, these current works focus on the basic task of predicting subsumptions between named concepts, without considering complex concepts defined with logical operators or complex logical relationships in evaluation.

In this paper, we propose Box$^2$EL, a novel OWL ontology embedding method targeting the semantics of $\mathcal{EL}^{++}$, which has been widely adopted in many real-life large-scale ontologies [16, 36].[4] To address the aforementioned limitations of existing approaches, we instead draw inspiration from BoxE [1] and represent both relations and concepts as boxes, while modeling interactions between concepts via a bumping mechanism. We not only demonstrate how Box$^2$EL overcomes the shortcomings of previous methods, but also prove that it is *sound*, i.e., faithfully captures the semantics of the underlying DL, which shows its theoretical correctness and supports interpretable inference for ontology completion. We evaluate our method in the two different inductive reasoning settings of concept subsumption prediction and role assertion (link) prediction, and on approximating deductive reasoning. Furthermore, we introduce a novel benchmark for subsumption prediction involving both named and complex concepts. Our results demonstrate that the theoretical advantages of our approach manifest themselves in practice and lead to state-of-the-art performance across a variety of datasets.

## 2 BACKGROUND AND RELATED WORK

### 2.1 Description Logic Ontologies

A DL ontology $O$ describes some domain of interest in terms of individuals, concepts and roles, where individuals correspond to objects in the domain, concepts represent sets of objects, and roles are binary relations between objects. We limit our discussion to the DL $\mathcal{EL}^{++}$ [3], which underpins the OWL 2 EL profile [21]. It is widely adopted since it contains many useful and important knowledge representation features, while allowing for reasoning in polynomial time. Given sets $\mathcal{N}_I$, $\mathcal{N}_C$ and $\mathcal{N}_R$ of, respectively, individual, concept, and role names, $\mathcal{EL}^{++}$ concepts are recursively defined as

$$\top \mid \bot \mid A \mid C \sqcap D \mid \exists r.C \mid \{a\}$$

where $\top$ is the top concept, $\bot$ is the bottom concept, $A \in \mathcal{N}_C$ is an atomic (or *named*) concept, $r \in \mathcal{N}_R$ is an atomic role, $a \in \mathcal{N}_I$ is an individual, and $C$ and $D$ are themselves (possibly complex) $\mathcal{EL}^{++}$ concepts. We say a concept is *complex* when it is constructed with a logical operator such as $\sqcap$ or $\exists$. An $\mathcal{EL}^{++}$ ontology $O$ consists of a TBox $\mathcal{T}$ and an ABox $\mathcal{A}$. The TBox consists of logical background knowledge in the form of concept subsumption axioms $C \sqsubseteq D$ and role inclusion axioms $r_1 \circ \cdots \circ r_k \sqsubseteq r$, while the ABox contains concrete data in the form of concept and role assertion axioms $C(a)$ and $r(a, b)$. Note that a relational fact from a KG in the form of an

RDF triple $(a, r, b)$ is equivalent to a role assertion axiom in the form of $r(a, b)$, and thus ontologies can be seen as extending KGs with more complex conceptual and logical information.

**Example 1.** The following ontology models a simple family domain:

$\mathcal{T} = \{$Father $\sqsubseteq$ Parent $\sqcap$ Male, Mother $\sqsubseteq$ Parent $\sqcap$ Female,

Child $\sqsubseteq$ $\exists$hasParent.Father, Child $\sqsubseteq$ $\exists$hasParent.Mother,

hasParent $\sqsubseteq$ relatedTo$\}$

$\mathcal{A} = \{$Father(Alex), Child(Bob), hasParent(Bob, Alex)$\}$

The TBox specifies that a father is a male parent, a mother is a female parent, every child has a father and a mother, and having a parent implies being related to that parent; the ABox states that Alex is a father, Bob is a child, and Alex is a parent of Bob.

Similarly to first order logic, the semantics of $\mathcal{EL}^{++}$ are defined in terms of *interpretations* that map individuals to elements, concept names to subsets, and role names to binary relations over some set $\Delta$, called the *interpretation domain*. An interpretation $\mathcal{I}$ that satisfies the semantics of every axiom in $O$ is called a *model* of $O$, denoted as $\mathcal{I} \models O$. See Baader et al. [3] for a formal definition.

*Remark.* $\mathcal{EL}^{++}$ also allows for so-called *concrete domains* (a.k.a. datatypes and values), which we do not consider in this paper. Technically, we work on the $\mathcal{ELHO}(\circ)^\perp$ subset of $\mathcal{EL}^{++}$.

### 2.2 Subsumption Inference

A central problem in DLs is to infer concept subsumptions from an ontology $O$. Such a subsumption can be either between two named concepts, or between a named concept and a complex concept. Classical reasoning algorithms leverage the logical information in $O$ to derive subsumptions that logically follow from the semantics; for example, we can infer from the ontology in Example 1 that Child $\sqsubseteq$ $\exists$relatedTo.Father. In contrast, *inductive* reasoning (also called *prediction*) aims to infer *probable* subsumptions from $O$. Note that when we limit ourselves to predicting subsumptions of the form $\{a\} \sqsubseteq \exists r.\{b\}$, which are equivalent to role assertion axioms $r(a, b)$, this is identical to the problem of *link prediction* in KGs (i.e., KG completion).

The majority of the existing prediction methods focus only on subsumptions between named concepts. Some approaches embed the formal semantics defined by the DL, while others focus on utilizing textual information such as concept labels and comments. For the former, please see Section 2.4. For the latter, OPA2Vec [35] and OWL2Vec* [8] use a Word2Vec model trained with local graph structure augmented corpora to embed the text for predicting subsumptions, while the recent method BERTSubs [7] fine-tunes a BERT model together with an attached classifier for predicting subsumptions involving both named concepts and complex concepts. Although these works consider a small part of the formal semantics such as the concept hierarchy as the context of a concept for augmenting prediction, they do not model the (complete) logical relationships of the ontology in the vector space. They are complementary to semantic ontology embedding methods including Box$^2$EL, but jointly embedding DL semantics and textual information is out of the scope of this paper.

---

[4]The logical constructors provided by $\mathcal{EL}^{++}$ are very common. While some OWL ontologies use more complicated features not supported by $\mathcal{EL}^{++}$, Box$^2$EL can still be used in that case to model the subset of axioms that fall into $\mathcal{EL}^{++}$.

## 2.3 Knowledge Graph Embeddings

KG embedding models such as TransE [6], DistMult [45], ComplEx [39] and BoxE [1] aim to solve the problem of completing KGs composed of purely relational facts, and can be thought of as modeling only the role assertion part of the ABox of an OWL ontology [40]. In particular, BoxE [1] also represents relations as boxes and adopts bump vectors, but it models relational facts alone, whereas we aim at much more complex DL ontologies with logical relationships involving concepts and roles.

Some KG embedding methods take background knowledge into account and are therefore related to ontology embedding techniques. However, these methods still focus only on modeling relational facts in a KG, using the background knowledge as constraints, and most of them only support logical rules concerning relations [13, 27, 28, 33, 41] or schemas in simple languages like RDF Schema [14, 43]. In contrast, ontology embedding methods including Box$^2$EL focus on OWL ontologies, which contain a large quantity of conceptual knowledge in the form of subsumptions and logical relationships. Furthermore, these methods can only be applied in the setting of link prediction, whereas ontology embeddings also allow predicting novel conceptual information or logical background knowledge itself.

## 2.4 Semantic Ontology Embeddings

Several ontology embedding methods for DL semantics have been proposed by learning geometric models. ELEm [22] is among the first to embed $\mathcal{EL}^{++}$, and EmEL$^{++}$[25] extends ELEm by considering role inclusion axioms. However, both methods represent concepts as high-dimensional balls, which have the disadvantage of not being closed under intersection. Our concept representation based on boxes has previously been used in the two recent methods BoxEL [44] and ELBE [31]. Mondal et al. [25] is the only other technique we are aware of that also models role inclusion axioms; the other methods consider only a smaller subset of $\mathcal{EL}^{++}$. All previous methods simply model roles (binary relations) by a single vector-based translation as in TransE [6], which fails to faithfully capture one-to-many, many-to-one, and many-to-many relations [23, 42]. We use box-based modeling in combination with bump vectors to address the above problem, achieving better performance in ontology completion.

Going beyond $\mathcal{EL}^{++}$, Özçep et al. [29] introduce a cone-based model for the more expressive DL $\mathcal{ALC}$; however, their contribution is mainly theoretical since they provide neither an implementation nor an evaluation. Embed2Reason [10] is another embedding approach for $\mathcal{ALC}$ ontologies based on quantum logic [5]. In contrast to our work, its focus is on ABox instead of subsumption reasoning. Finally, it is worth mentioning that in comparison with all these DL embedding works, we conduct a more thorough evaluation with our new benchmark for predicting subsumptions between not only named concepts, but also named concepts and complex concepts involving logical operators.

## 3 METHOD

In order to perform inductive reasoning over an $\mathcal{EL}^{++}$ ontology $\mathcal{O}$ with signature $\Sigma = (\mathcal{N}_C, \mathcal{N}_R, \mathcal{N}_I)$, we follow the general framework of Kulmanov et al. [22] and learn embeddings that correspond to geometric models of $\mathcal{O}$; that is, (logical) models with interpretation domain $\Delta = \mathbb{R}^n$. We now specify how Box$^2$EL maps concepts, individuals, and roles to the embedding space $\mathbb{R}^n$ and describe the loss functions that encode the axioms of $\mathcal{O}$.

### 3.1 Geometric Representation

*Concepts and individuals.* We follow recent work [31, 44] and represent concepts as $n$-dimensional *boxes*, i.e., *axis-aligned hyperrectangles*. This representation has several advantages over the alternative based on $n$-dimensional *balls*, such as closure under concept intersection, and has been shown to work well in practice. Formally, we associate with every concept $C \in \mathcal{N}_C$ two vectors $l_C \in \mathbb{R}^n$ and $u_C \in \mathbb{R}^n$ such that $l_C \leq u_C$, where $\leq$ is applied element-wise. These vectors form the *lower* and *upper* corner of the box of $C$, i.e., $\text{Box}(C) = \{ x \in \mathbb{R}^n \mid l_C \leq x \leq u_C \}$. The *center* of $\text{Box}(C)$ is given by $(l_C + u_C)/2$, and its *offset* is $(u_C - l_C)/2$.

We represent individuals $a \in \mathcal{N}_I$ as *points* $e_a \in \mathbb{R}^n$ in the embedding space. Nominals (concepts of the form $\{a\}$), are then formally mapped to boxes with volume 0, i.e., $l_{\{a\}} = u_{\{a\}} = e_a$.

*Roles.* While most existing $\mathcal{EL}^{++}$ embedding models represent roles (binary relations) via simple translations in the form of TransE [6], we instead follow the idea of the BoxE KG embedding model [1]. That is, we associate every role $r \in \mathcal{N}_R$ with a *head box* $\text{Head}(r)$ and a *tail box* $\text{Tail}(r)$. Intuitively, every point in the head box is related via $r$ to every point in the tail box. This representation is made more expressive by introducing bump vectors $\text{Bump}(C)$ for every atomic concept $C$, which model interactions between concepts by dynamically moving the embeddings of concepts related via a role $r$. An axiom of the form $C \sqsubseteq \exists r.D$ is then considered to hold if

$$\begin{aligned} \text{Box}(C) \oplus \text{Bump}(D) &\subseteq \text{Head}(r) \quad \text{and} \\ \text{Box}(D) \oplus \text{Bump}(C) &\subseteq \text{Tail}(r), \end{aligned} \tag{1}$$

where $\oplus$ denotes the operation of translating a box along a bump vector.

**Example 2.** Figure 1 illustrates 2-dimensional Box$^2$EL embeddings that form a logical model of the TBox in Example 1, since, e.g., Box(Father) $\subseteq$ Box(Parent) $\cap$ Box(Male) and Equation (1) holds for all relevant axioms. We will formalise the logical geometric model associated with a set of Box$^2$EL embeddings in Theorem 1. Note also that the embeddings furthermore imply the subsumption Child $\sqsubseteq \exists$relatedTo.Father (again by Equation (1)), illustrating their utility for approximate reasoning.

*Expressiveness.* The previous example illustrates the expressive power of Box$^2$EL. Representing roles as head and tail boxes allows modeling one-to-many relationships such as hasParent faithfully, in contrast to previous approaches that employ a TransE-based role representation. Moreover, role inclusion axioms such as hasParent $\sqsubseteq$ relatedTo can be represented naturally via inclusion constraints on the relevant head and tail boxes. In contrast, previous methods either do not consider role inclusion axioms at all or only rudimentarily approximate them by forcing the embeddings of the involved roles to be similar [25].

*Model complexity.* Box$^2$EL requires $2n|\mathcal{N}_C| + n|\mathcal{N}_I|$ parameters to store the lower and upper corners of the box embeddings of concepts and points associated with individuals. In order to represent

**Figure 1: An illustration of Box$^2$EL embeddings. Striped boxes represent concept embeddings, whereas role embeddings are shaded blue and labelled as $r^h$ or $r^t$ for the head or tail box of $r$, respectively. Bump vectors are drawn as arrows and labelled with the corresponding concept. The illustrated embeddings form a logical model of the TBox in Example 1.**

the head and tail boxes for every relation and a bump vector per concept and individual, the model needs $4n|\mathcal{N}_R| + n(|\mathcal{N}_C| + |\mathcal{N}_I|)$ additional parameters. Therefore, the total space complexity of Box$^2$EL is in $O\Big(n\big(3|\mathcal{N}_C| + 2|\mathcal{N}_I| + 4|\mathcal{N}_R|\big)\Big)$.

## 3.2 Training Procedure

In order to learn embeddings for $O$, we first unify its ABox and TBox with the transformation rules

$$C(a) \rightsquigarrow \{a\} \sqsubseteq C$$
$$r(a,b) \rightsquigarrow \{a\} \sqsubseteq \exists r.\{b\}$$

and then normalize the axioms using a standard procedure (e.g., Baader et al. [3]). We introduce separate loss functions for axioms in each normal form, which intuitively ensure that the learned embeddings adhere to the semantics of $O$. Finally, we minimize the sum of all loss terms $\mathcal{L}(O)$ via mini-batch gradient descent.

Our loss functions are based on the distance-based loss formulation in [22, 31] and aim to minimize the element-wise distance between the embeddings of related concepts. Given two arbitrary boxes $A$ and $B$, this element-wise distance is computed as

$$\boldsymbol{d}(A,B) = |\boldsymbol{c}(A) - \boldsymbol{c}(B)| - \boldsymbol{o}(A) - \boldsymbol{o}(B),$$

where $\boldsymbol{c}(\cdot)$ and $\boldsymbol{o}(\cdot)$ denote the center and offset of a box, respectively. Note that for nominals $\{a\}$ we have that $\boldsymbol{c}(\text{Box}(\{a\})) = \boldsymbol{e}_a$ and $\boldsymbol{o}(\text{Box}(\{a\})) = \boldsymbol{0}$.

*Generic inclusion loss.* Before defining loss functions for the different $\mathcal{EL}^{++}$ normal forms, we first introduce a generic inclusion loss $\mathcal{L}_\subseteq(A,B)$. It encourages box $A$ to be contained in box $B$ and is defined as

$$\mathcal{L}_\subseteq(A,B) = \begin{cases} \|\max\{\boldsymbol{0}, \boldsymbol{d}(A,B) + 2\boldsymbol{o}(A) - \gamma\}\| & \text{if } B \neq \emptyset \\ \max\{0, \boldsymbol{o}(A)_1 + 1\} & \text{otherwise,} \end{cases}$$

where $\gamma$ is a margin hyperparameter. If $\mathcal{L}_\subseteq(A,B) = 0$, either $A$ lies within $\gamma$-distance of $B$ in each dimension, or both $A$ and $B$ are empty.

We next introduce each normal form and the corresponding loss function. Note that all concepts in the normal forms below are atomic concepts or nominals (and not complex).

*First normal form (NF1).* For an NF1 axiom of the form $C \sqsubseteq D$, the learned embeddings need to satisfy $\text{Box}(C) \subseteq \text{Box}(D)$, corresponding to the semantics of concept inclusion. Therefore, we define the loss for NF1 as simply the inclusion loss:

$$\mathcal{L}_1(C,D) = \mathcal{L}_\subseteq(\text{Box}(C), \text{Box}(D)).$$

*Second normal form (NF2).* For an NF2 axiom of the form $C \sqcap D \sqsubseteq E$, we similarly require that the intersection of the boxes of $C$ and $D$ is within the box of $E$. The intersection of $\text{Box}(C)$ and $\text{Box}(D)$ is itself a box with lower corner $\max\{\boldsymbol{l}_C, \boldsymbol{l}_D\}$ and upper corner $\min\{\boldsymbol{u}_C, \boldsymbol{u}_D\}$, where max and min are applied element-wise. We thus have

$$\mathcal{L}_2(C,D,E) = \mathcal{L}_\subseteq\Big(\text{Box}(C) \cap \text{Box}(D), \text{Box}(E)\Big).$$

However, this formulation is problematic since it can be easily minimized to 0 by setting $\text{Box}(C)$ and $\text{Box}(D)$ to be disjoint. While disjoint embeddings for $C$ and $D$ would technically not violate the semantics, usually an axiom of the form $C \sqcap D \sqsubseteq \bot$ would have been used directly if it had been the intention that $C$ and $D$ should be disjoint. Therefore, we add the additional term

$$\|\max\{\boldsymbol{0}, \max\{\boldsymbol{l}_C, \boldsymbol{l}_D\} - \min\{\boldsymbol{u}_C, \boldsymbol{u}_D\}\}\|$$

to the loss, which intuitively encourages $\text{Box}(C) \cap \text{Box}(D)$ to be non-empty by making all elements of its offset vector positive.

*Third normal form (NF3).* For an NF3 axiom of the form $C \sqsubseteq \exists r.D$, the embeddings should satisfy $\text{Box}(C) + \text{Bump}(D) \subseteq \text{Head}(r)$ and $\text{Box}(D) + \text{Bump}(C) \subseteq \text{Tail}(r)$. This is captured by the following loss function:

$$\mathcal{L}_3(C,r,D) = \frac{1}{2}\Big(\mathcal{L}_\subseteq(\text{Box}(C) + \text{Bump}(D), \text{Head}(r))$$
$$+ \mathcal{L}_\subseteq(\text{Box}(D) + \text{Bump}(C), \text{Tail}(r))\Big).$$

If $\text{Box}(D)$ is empty, we furthermore add the term $\mathcal{L}_\subseteq(\text{Box}(C), \emptyset)$ to also make $\text{Box}(C)$ empty.

*Fourth normal form (NF4).* For an NF4 axiom of the form $\exists r.C \sqsubseteq D$, we need to ensure that all points in the embedding space that are connected to $C$ via role $r$ are contained in $\text{Box}(D)$. It can be seen from our geometric representation that the set of these points is contained in the set $\text{Head}(r) - \text{Bump}(C)$. We therefore define the loss for the fourth normal form as

$$\mathcal{L}_4(r, C, D) = \mathcal{L}_{\subseteq}(\text{Head}(r) - \text{Bump}(C), \text{Box}(D)).$$

*Fifth normal form (NF5).* Axioms of the fifth normal form $C \sqcap D \sqsubseteq \bot$ state that concepts $C$ and $D$ are disjoint. Our corresponding loss function penalizes embeddings for which the element-wise distance is not greater than 0 (within a margin of $\gamma$) and is defined as

$$\mathcal{L}_5(C, D) = \|\max\{\mathbf{0}, -(\boldsymbol{d}(\text{Box}(C), \text{Box}(D)) + \gamma)\}\|.$$

*Role inclusion axioms.* After normalization, role inclusion axioms are either of the form $r \sqsubseteq s$ or $r_1 \circ r_2 \sqsubseteq s$. For the first case, we define the loss function

$$\mathcal{L}_6(r, s) = \frac{1}{2}\Big(\mathcal{L}_{\subseteq}(\text{Head}(r), \text{Head}(s)) + \mathcal{L}_{\subseteq}(\text{Tail}(r), \text{Tail}(s))\Big),$$

which intuitively makes any embeddings related via role $r$ also related via $s$. Similarly, in the second case we have the loss

$$\mathcal{L}_7(r_1, r_2, s) = \frac{1}{2}\Big(\mathcal{L}_{\subseteq}(\text{Head}(r_1), \text{Head}(s)) + \mathcal{L}_{\subseteq}(\text{Tail}(r_2), \text{Tail}(s))\Big).$$

*Regularization.* In order to prevent our expressive role representation from overfitting, we furthermore $L^2$-regularize the bump vectors by adding the regularization term

$$\lambda \sum_{C \in \mathcal{N}_C \cup \mathcal{N}_I} \|\text{Bump}(C)\|,$$

where $\lambda$ is a hyperparameter.

*Negative sampling.* In addition to the above loss functions, we also employ *negative sampling* to further improve the quality of the learned embeddings. We follow previous work [22, 31] and generate negative samples for axioms of the form $C \sqsubseteq \exists r.D$ by replacing either $C$ or $D$ with a randomly selected different concept, similar to negative sampling in KGs [6]. For every NF3 axiom we generate a new set of $\omega \geq 1$ negative samples $C \not\sqsubseteq \exists r.D$ in every epoch, for which we optimize the loss

$$\mathcal{L}_{\not\sqsubseteq}(C, r, D) = (\delta - \mu(\text{Box}(C) + \text{Bump}(D), \text{Head}(r)))^2$$
$$+ (\delta - \mu(\text{Box}(D) + \text{Bump}(C), \text{Tail}(r)))^2,$$

where $\mu(A, B) = \|\max\{\mathbf{0}, \boldsymbol{d}(A, B) + \gamma\}\|$ is the minimal distance between any two points in $A$ and $B$ and $\delta > 0$ is a hyperparameter that controls how unlikely the negative samples are made by the model.

As in the case of KGs, the above procedure may occasionally generate false negative samples, i.e., negative axioms that actually do follow from $O$. However, on average it will predominantly produce true negatives, which we find to empirically improve the learned embeddings.

## 3.3 Soundness

We now show that the embeddings learned by Box$^2$EL indeed correspond to a logical geometric model of the given ontology $O$.

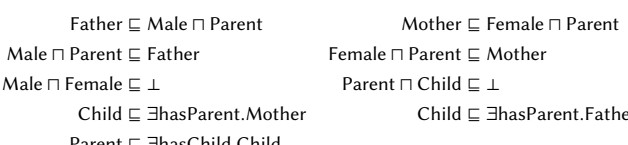

Father $\sqsubseteq$ Male $\sqcap$ Parent     Mother $\sqsubseteq$ Female $\sqcap$ Parent

Male $\sqcap$ Parent $\sqsubseteq$ Father     Female $\sqcap$ Parent $\sqsubseteq$ Mother

Male $\sqcap$ Female $\sqsubseteq$ $\bot$     Parent $\sqcap$ Child $\sqsubseteq$ $\bot$

Child $\sqsubseteq$ $\exists$hasParent.Mother     Child $\sqsubseteq$ $\exists$hasParent.Father

Parent $\sqsubseteq$ $\exists$hasChild.Child

**Figure 2: Proof of concept ontology.**

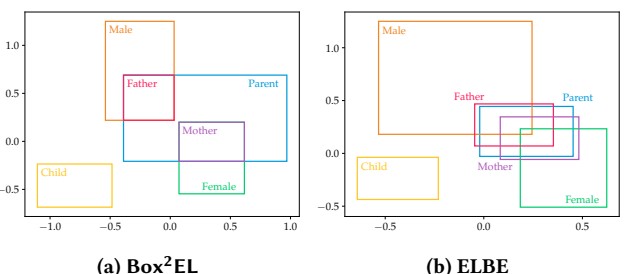

(a) Box$^2$EL          (b) ELBE

**Figure 3: Visualization of the embeddings learned by Box$^2$EL (left) and ELBE (right) for the proof-of-concept ontology. While Box$^2$EL can accurately represent the axioms in the ontology, the limitations of TransE as a model for roles prevent ELBE from learning correct embeddings.**

THEOREM 1 (SOUNDNESS). *Let $O = (\mathcal{T}, \mathcal{A})$ be an $\mathcal{EL}^{++}$ ontology. If $\gamma \leq 0$ and there exist Box$^2$EL embeddings in $\mathbb{R}^n$ such that $\mathcal{L}(O) = 0$, then these embeddings are a model of $O$.*

PROOF SKETCH. We construct a geometric interpretation $\mathcal{I}$ of $O$ by interpreting every individual as its associated point, every concept as the set of points in its associated box, and every role as the Cartesian product of the set of points in its head box and the set of points in its tail box. Since $\mathcal{L}(O) = 0$, we have that $\mathcal{I}$ is a model of $O$. □

A formal proof of Theorem 1 is given in Appendix A. While our optimization procedure might not achieve a loss of 0 in practice, the importance of Theorem 1 is that it demonstrates that the learned embeddings converge to a semantically meaningful representation of the ontology in which all of its axioms are satisfied, i.e., a model of $O$. The embeddings therefore indeed encode the axioms in $O$ and are thus useful to perform inductive or approximate reasoning.

## 4 EVALUATION

We first validate our model and demonstrate its expressiveness on a proof of concept ontology. We then evaluate Box$^2$EL on three tasks: general subsumption prediction, link (role assertion) prediction, and approximating deductive reasoning. Furthermore, we conduct a variety of ablation studies whose results are shown in Appendix C. Our implementation is based on PyTorch [30], and we use the jcel reasoner [24] to transform ontologies into normal form axioms.

## 4.1 Proof of Concept: Family Ontology

We visualize the embeddings learned by Box$^2$EL for a proof of concept family ontology given in Figure 2. To this end, we train

**Table 1: Overall subsumption prediction results combined across normal forms.**

| | Model | H@1 | H@10 | H@100 | Med | MRR | MR | AUC |
|---|---|---|---|---|---|---|---|---|
| GALEN | ELEm | 0.01 | 0.12 | 0.29 | 1662 | 0.05 | 5153 | 0.78 |
| | EmEL$^{++}$ | 0.01 | 0.11 | 0.24 | 2295 | 0.05 | 5486 | 0.76 |
| | BoxEL | 0.00 | 0.03 | 0.16 | 4785 | 0.01 | 7163 | 0.69 |
| | ELBE | 0.02 | 0.14 | 0.27 | 1865 | 0.06 | 5303 | 0.77 |
| | Box$^2$EL | **0.05** | **0.20** | **0.35** | **669** | **0.10** | **4375** | **0.81** |
| GO | ELEm | 0.03 | **0.24** | 0.43 | 272 | 0.09 | 6204 | 0.86 |
| | EmEL$^{++}$ | 0.03 | 0.23 | 0.38 | 597 | 0.09 | 6710 | 0.85 |
| | BoxEL | 0.01 | 0.06 | 0.08 | 8443 | 0.03 | 14905 | 0.68 |
| | ELBE | 0.01 | 0.10 | 0.22 | 1838 | 0.04 | 6986 | 0.85 |
| | Box$^2$EL | **0.04** | 0.23 | **0.59** | **48** | **0.10** | **3248** | **0.93** |
| Anatomy | ELEm | 0.10 | 0.40 | 0.64 | 22 | 0.19 | 6464 | 0.94 |
| | EmEL$^{++}$ | 0.11 | 0.36 | 0.57 | 36 | 0.19 | 8472 | 0.92 |
| | BoxEL | 0.03 | 0.12 | 0.28 | 1151 | 0.06 | 10916 | 0.90 |
| | ELBE | 0.04 | 0.36 | 0.63 | 29 | 0.15 | 5400 | 0.95 |
| | Box$^2$EL | **0.16** | **0.47** | **0.70** | **13** | **0.26** | **2675** | **0.97** |

**Table 2: Detailed subsumption prediction results on the GALEN ontology.**

| | Model | H@1 | H@10 | H@100 | Med | MRR | MR | AUC |
|---|---|---|---|---|---|---|---|---|
| NF1 | ELEm | 0.01 | 0.16 | 0.40 | 430 | 0.06 | 3568 | 0.85 |
| | EmEL$^{++}$ | 0.02 | 0.16 | 0.37 | 632 | 0.06 | 3765 | 0.84 |
| | BoxEL | 0.00 | 0.00 | 0.05 | 3715 | 0.00 | 5727 | 0.75 |
| | ELBE | **0.03** | 0.24 | 0.47 | 138 | 0.10 | **2444** | **0.89** |
| | Box$^2$EL | **0.03** | **0.30** | **0.51** | **91** | **0.12** | 2632 | **0.89** |
| NF2 | ELEm | 0.01 | 0.07 | 0.17 | 5106 | 0.03 | 7432 | 0.68 |
| | EmEL$^{++}$ | 0.01 | 0.07 | 0.15 | 5750 | 0.03 | 7767 | 0.66 |
| | BoxEL | 0.00 | 0.00 | 0.00 | 11358 | 0.00 | 11605 | 0.50 |
| | ELBE | 0.03 | 0.06 | 0.11 | 6476 | 0.04 | 8068 | 0.65 |
| | Box$^2$EL | **0.06** | **0.15** | **0.28** | **2149** | **0.09** | **6265** | **0.73** |
| NF3 | ELEm | 0.02 | 0.14 | 0.28 | 1479 | 0.05 | 4831 | 0.79 |
| | EmEL$^{++}$ | 0.02 | 0.11 | 0.22 | 2240 | 0.05 | 5348 | 0.77 |
| | BoxEL | 0.00 | 0.02 | 0.08 | 7239 | 0.01 | 8615 | 0.63 |
| | ELBE | 0.03 | 0.14 | 0.25 | 2154 | 0.07 | 5072 | 0.78 |
| | Box$^2$EL | **0.08** | **0.19** | **0.32** | **635** | **0.12** | **3798** | **0.84** |
| NF4 | ELEm | 0.00 | 0.05 | 0.18 | 3855 | 0.02 | 6793 | 0.71 |
| | EmEL$^{++}$ | 0.00 | 0.04 | 0.12 | 4458 | 0.01 | 7020 | 0.70 |
| | BoxEL | 0.00 | **0.15** | **0.69** | **47** | **0.04** | **2667** | **0.89** |
| | ELBE | 0.00 | 0.03 | 0.07 | 7563 | 0.01 | 8884 | 0.62 |
| | Box$^2$EL | 0.00 | 0.06 | 0.15 | 4364 | 0.02 | 7266 | 0.69 |

Box$^2$EL with an embedding dimensionality of $n = 2$, a margin of $\gamma = 0$, regularization strength $\lambda = 1$, and no negative sampling. We also train embeddings for ELBE [31], a comparable $\mathcal{EL}^{++}$ ontology embedding model that similarly uses boxes to represent concepts, but interprets roles as translations. In order to ensure the volume of the learned embeddings is large enough for plotting, we add the following visualization loss term to the objective function of both models:

$$\mathcal{L}_V = \frac{1}{n|\mathcal{N}_C|} \sum_{C \in \mathcal{N}_C} \sum_{i=1}^{n} \max\{0, 0.2 - \boldsymbol{o}(\text{Box}(C))_i\}.$$

The learned concept embeddings of both models are depicted in Figure 3. We see that Box$^2$EL is able to successfully learn embeddings that align with the axioms in the ontology. In particular, the embeddings fulfill all disjointness axioms and correctly represent the relationships between the concepts Father, Male, Mother, Female, and Parent.

In contrast, we find that the embeddings learned by ELBE violate several of the axioms in the ontology. This is due to the inability of the underlying TransE model to correctly represent one-to-many relationships: because the ontology contains the axioms Child ⊑ ∃hasParent.Mother as well as Child ⊑ ∃hasParent.Father, the model is forced to let the embeddings of Mother and Father overlap.

## 4.2 General Subsumption Prediction

We next evaluate Box$^2$EL on general subsumption prediction for inductive reasoning. In contrast to previous work [25, 44], we not only consider subsumptions between atomic (named) concepts, but also the more challenging task of predicting subsumptions between atomic concepts and complex concepts.

*Benchmark.* We introduce a novel benchmark based on three biomedical ontologies GALEN [32], Gene Ontology (GO) [2] and Anatomy (a.k.a. Uberon) [26], which are also adopted in the previous DL embedding works [25, 44]. For each ontology, our benchmark consists of axioms split into training (80%), validation (10%), and testing (10%) sets for each normal form. This enables us to

evaluate ontology embedding models on subsumption prediction between atomic concepts (NF1), atomic concepts and conjunctions (NF2), and atomic concepts and existential restrictions (NF3 and NF4). We report statistics on the sizes of these ontologies, including the numbers of axioms of different forms, in Table 7 in the appendix.

*Baselines.* We compare Box$^2$EL with the state-of-the-art ontology embedding methods ELEm [22], EmEL$^{++}$ [25], BoxEL [44], and ELBE [31]. We do not consider any traditional KG embedding methods in our experiments, since they have been shown to be considerably outperformed by ontology embedding methods [25, 44] and are not applicable in the setting of complex concepts.

*Evaluation protocol.* To evaluate the subsumption prediction performance, we follow the literature [25, 44] and report a variety of ranking-based metrics on the testing set. Given a test axiom in some normal form, we generate a set of candidate predictions by replacing the atomic side of the subsumption with all the atomic concepts in $\mathcal{N}_C$. We then rank all candidate predictions by a score based on the distance between the embeddings of the concepts of the subsumption (for details see Appendix D) and record the rank of the true axiom. We report the standard metrics Hits@$k$ (H@$k$), where $k \in \{1, 10, 100\}$, the median rank (Med), the mean reciprocal rank (MRR), the mean rank (MR), and the area under the ROC curve (AUC). These metrics are computed for the axioms in each normal form individually, as well as combined across normal forms.

*Experimental protocol.* The embeddings are optimized with Adam [20] for a maximum of 10,000 epochs. All hyperparameters are described in detail in Appendix E. We evaluate the models on a fraction of the validation set every 100 epochs and choose the embeddings that achieve the best performance for final evaluation on the testing set. The results we report are averages across 5 runs with different random seeds, which we provide in our implementation for

**Table 3: Detailed subsumption prediction results on the GO ontology.**

| | Model | H@1 | H@10 | H@100 | Med | MRR | MR | AUC |
|---|---|---|---|---|---|---|---|---|
| NF1 | ELEm | 0.01 | 0.13 | 0.35 | 590 | 0.05 | 6433 | 0.86 |
| | EmEL++ | 0.01 | 0.12 | 0.30 | 1023 | 0.05 | 6709 | 0.85 |
| | BoxEL | 0.00 | 0.01 | 0.05 | 5374 | 0.00 | 13413 | 0.71 |
| | ELBE | 0.01 | 0.10 | 0.24 | 1156 | 0.04 | 5657 | 0.88 |
| | Box$^2$EL | **0.03** | **0.17** | **0.58** | **58** | **0.08** | **2686** | **0.94** |
| NF2 | ELEm | 0.12 | 0.49 | 0.63 | 11 | 0.24 | 4508 | 0.90 |
| | EmEL++ | 0.11 | 0.44 | 0.55 | 23 | 0.21 | 5169 | 0.89 |
| | BoxEL | 0.00 | 0.00 | 0.00 | 22882 | 0.00 | 23007 | 0.50 |
| | ELBE | 0.01 | 0.05 | 0.09 | 6456 | 0.02 | 9421 | 0.80 |
| | Box$^2$EL | **0.18** | **0.58** | **0.75** | **6** | **0.31** | **2104** | **0.95** |
| NF3 | ELEm | **0.06** | **0.40** | 0.52 | **54** | **0.15** | 6292 | 0.86 |
| | EmEL++ | 0.05 | 0.39 | 0.48 | 210 | **0.15** | 7788 | 0.83 |
| | BoxEL | 0.00 | 0.00 | 0.00 | 17027 | 0.00 | 18947 | 0.59 |
| | ELBE | 0.02 | 0.15 | 0.30 | 959 | 0.07 | 7131 | 0.84 |
| | Box$^2$EL | 0.00 | 0.18 | **0.53** | 79 | 0.05 | **5042** | **0.89** |
| NF4 | ELEm | 0.01 | 0.49 | 0.60 | **12** | 0.12 | 6272 | 0.86 |
| | EmEL++ | 0.01 | 0.49 | 0.58 | **12** | 0.13 | 6442 | 0.86 |
| | BoxEL | **0.09** | **0.54** | 0.54 | 2215 | **0.28** | 9673 | 0.79 |
| | ELBE | 0.00 | 0.07 | 0.12 | 9049 | 0.02 | 12868 | 0.72 |
| | Box$^2$EL | 0.00 | 0.37 | **0.64** | 20 | 0.08 | **4989** | **0.89** |

**Table 4: Detailed subsumption prediction results on the Anatomy ontology.**

| | Model | H@1 | H@10 | H@100 | Med | MRR | MR | AUC |
|---|---|---|---|---|---|---|---|---|
| NF1 | ELEm | 0.07 | 0.30 | 0.57 | 43 | 0.14 | 9059 | 0.91 |
| | EmEL++ | **0.08** | 0.29 | 0.53 | 60 | 0.14 | 10414 | 0.90 |
| | BoxEL | 0.01 | 0.05 | 0.16 | 1828 | 0.03 | 9597 | 0.91 |
| | ELBE | 0.05 | 0.24 | 0.55 | 68 | 0.11 | 5177 | 0.95 |
| | Box$^2$EL | 0.07 | **0.34** | **0.65** | **27** | **0.15** | **2894** | **0.97** |
| NF2 | ELEm | 0.03 | 0.18 | 0.42 | 394 | 0.08 | 11592 | 0.89 |
| | EmEL++ | 0.03 | 0.18 | 0.35 | 1291 | 0.08 | 15759 | 0.85 |
| | BoxEL | 0.00 | 0.00 | 0.00 | 17607 | 0.00 | 26872 | 0.75 |
| | ELBE | 0.02 | 0.11 | 0.26 | 1394 | 0.05 | 4885 | 0.96 |
| | Box$^2$EL | **0.16** | **0.41** | **0.64** | **26** | **0.24** | **1928** | **0.98** |
| NF3 | ELEm | 0.12 | 0.47 | 0.69 | 13 | 0.23 | 4686 | 0.96 |
| | EmEL++ | 0.13 | 0.42 | 0.60 | 23 | 0.23 | 7097 | 0.93 |
| | BoxEL | 0.04 | 0.17 | 0.36 | 567 | 0.08 | 11095 | 0.90 |
| | ELBE | 0.04 | 0.44 | 0.70 | 16 | 0.18 | 5408 | 0.95 |
| | Box$^2$EL | **0.21** | **0.56** | **0.75** | **7** | **0.33** | **2466** | **0.98** |
| NF4 | ELEm | 0.00 | 0.03 | **0.23** | 813 | **0.01** | 10230 | 0.91 |
| | EmEL++ | 0.00 | 0.02 | 0.17 | 1470 | **0.01** | 10951 | 0.90 |
| | BoxEL | 0.00 | 0.00 | 0.00 | 38942 | 0.00 | 41283 | 0.61 |
| | ELBE | 0.00 | 0.02 | 0.06 | 6261 | **0.01** | 15187 | 0.86 |
| | Box$^2$EL | 0.00 | **0.05** | 0.14 | 3065 | **0.01** | **8366** | **0.92** |

reproducibility. All experiments were conducted with an NVIDIA Quadro RTX 8000 GPU (48G).

*Results.* The results on all the testing axioms (combined across all normal forms) are reported in Table 1. For detailed results on testing axioms of each normal form, see Tables 2 to 4. We first observe that our model Box$^2$EL consistently outperforms all the baselines on all datasets, often with significant performance gains. For example, the median rank (MR) of Box$^2$EL is around 60% lower than the second best-performing method on GALEN, more than 80% lower on GO, and more than 40% lower on Anatomy. Among the baseline methods, results are similar; interestingly, ELEm generally performs best, in contrast to previous benchmarks.

From the detailed results, we observe that the novel role representation of Box$^2$EL not only generally improves prediction performance for NF3 axioms, which contain roles, but also for NF1 and NF2 axioms. This can be explained by the fact that the different normal forms are used to optimize the *same* embeddings; i.e., if Box$^2$EL can better represent an axiom of the form $C \sqsubseteq \exists r.D$, it will learn better embeddings for $C$ and $D$, therefore also improving prediction quality for NF1 and NF2 axioms involving $C$ and/or $D$. There is no clear trend which axioms are the easiest to predict; on GALEN, the models generally perform better on NF1 axioms involving only atomic concepts, whereas on GO and Anatomy they perform similarly well on axioms involving complex concepts.

### 4.3 Link Prediction

We next evaluate our model on the task of link prediction, i.e., predicting role assertions of the form $r(a, b)$, which is implemented by predicting subsumptions of the form $\{a\} \sqsubseteq \exists r.\{b\}$.

*Datasets.* We consider a real-world protein-protein interaction (PPI) prediction task introduced in [22]. They provide two ontologies for human and yeast organisms, constructed by combining the STRING database of PPIs [38] with the Gene Ontology [2]. The proteins and their interactions recorded in STRING constitute the ABox, while GO acts as the TBox, and additional axioms modeling the association of proteins with their biological functions are added to the ontology. The task is to predict missing subsumptions of the form $\{P_1\} \sqsubseteq \exists interacts.\{P_2\}$, where $P_1$ and $P_2$ represent two proteins.

*Baselines.* We also consider ELEm [22], EmEL$^{++}$ [25], BoxEL [44], and ELBE [31] for the baselines as in general subsumption prediction, and report the relevant best results from their original papers.

*Evaluation and experimental protocol.* In order to evaluate our method, we use the 80%/10%/10% training, validation, and testing split of the PPI data provided by [22]. We compute the same ranking-based metrics as in subsumption prediction, both in the standard and filtered fashion, in which any true candidate predictions except for the target axiom to predict are first removed from the set of all candidate predictions before computing the ranks. The experimental protocol is the same as in subsumption prediction.

*Results.* Table 5 lists the results of Box$^2$EL and the baseline methods on the yeast and human PPI prediction datasets. Box$^2$EL outperforms all the baselines, with significant performance gains for both datasets on most of the metrics including filtered hits and mean rank. The AUC and AUC (F) values are all very close to the maximum value 1.0, but Box$^2$EL still improves the state-of-the-art on Yeast, and ties ELBE on Human. All these results demonstrate the effectiveness of Box$^2$EL in role assertion prediction for ontologies with an ABox. The comparatively stronger results than ELBE and BoxEL, which share the same concept representation as our model, once again highlight the positive impact of our novel approach of representing semantics of roles with boxes and bump vectors.

**Table 5: PPI prediction results on the yeast and human datasets. Columns annotated with (F) contain filtered metrics, other columns contain raw metrics. The results for BoxEL are from [44]; all other baseline results are from [31].**

|  | Model | H@10 | H@10 (F) | H@100 | H@100 (F) | MR | MR (F) | AUC | AUC (F) |
|---|---|---|---|---|---|---|---|---|---|
| Yeast | ELEm | 0.10 | 0.23 | 0.50 | 0.75 | 247 | 187 | 0.96 | 0.97 |
|  | EmEL$^{++}$ | 0.08 | 0.17 | 0.48 | 0.65 | 336 | 291 | 0.94 | 0.95 |
|  | BoxEL | 0.09 | 0.20 | 0.52 | 0.73 | 423 | 379 | 0.93 | 0.94 |
|  | ELBE | **0.11** | 0.26 | 0.57 | 0.77 | 201 | 154 | 0.96 | 0.97 |
|  | Box$^2$EL | **0.11** | **0.33** | **0.64** | **0.87** | **168** | **118** | **0.97** | **0.98** |
| Human | ELEm | **0.09** | 0.22 | 0.43 | 0.70 | 658 | 572 | 0.96 | 0.96 |
|  | EmEL$^{++}$ | 0.04 | 0.13 | 0.38 | 0.56 | 772 | 700 | 0.95 | 0.95 |
|  | BoxEL | 0.07 | 0.10 | 0.42 | 0.63 | 1574 | 1530 | 0.93 | 0.93 |
|  | ELBE | **0.09** | 0.22 | 0.49 | 0.72 | 434 | 362 | 0.97 | **0.98** |
|  | Box$^2$EL | **0.09** | **0.28** | **0.55** | **0.83** | 343 | 269 | **0.98** | **0.98** |

**Table 6: Deductive reasoning results on GALEN, GO, and Anatomy.**

|  | Model | H@1 | H@10 | H@100 | Med | MRR | MR | AUC |
|---|---|---|---|---|---|---|---|---|
| GALEN | ELEm | 0.00 | 0.04 | 0.20 | 1807 | 0.01 | 4405 | 0.81 |
|  | EmEL$^{++}$ | 0.00 | 0.04 | 0.18 | 2049 | 0.01 | 4634 | 0.81 |
|  | BoxEL | 0.00 | 0.00 | 0.01 | 6906 | 0.00 | 7925 | 0.67 |
|  | ELBE | 0.00 | 0.06 | 0.16 | 1785 | 0.02 | 3974 | 0.84 |
|  | Box$^2$EL | **0.01** | **0.09** | **0.24** | **1003** | **0.03** | **2833** | **0.88** |
| GO | ELEm | 0.00 | 0.04 | 0.22 | 1629 | 0.02 | 7377 | 0.84 |
|  | EmEL$^{++}$ | 0.00 | 0.04 | 0.19 | 1346 | 0.01 | 6557 | 0.86 |
|  | BoxEL | 0.00 | 0.00 | 0.13 | 1085 | 0.00 | 5359 | 0.88 |
|  | ELBE | 0.00 | 0.06 | 0.21 | 935 | 0.02 | 3846 | 0.92 |
|  | Box$^2$EL | 0.00 | **0.08** | **0.49** | **107** | **0.04** | **1689** | **0.96** |
| Anatomy | ELEm | 0.00 | 0.07 | 0.28 | 901 | 0.02 | 7958 | 0.93 |
|  | EmEL$^{++}$ | 0.00 | 0.07 | 0.26 | 1576 | 0.02 | 10976 | 0.90 |
|  | BoxEL | **0.01** | **0.10** | 0.24 | 838 | **0.04** | 9156 | 0.92 |
|  | ELBE | 0.00 | 0.08 | 0.32 | 336 | 0.03 | 2312 | 0.98 |
|  | Box$^2$EL | **0.01** | 0.09 | **0.44** | **152** | **0.04** | **1599** | **0.99** |

## 4.4 Approximating Deductive Reasoning

We finally investigate how well our model can approximate *deductive reasoning*, i.e., infer subsumptions that are logical consequences of the axioms in the ontology.

*Experimental setup.* We again consider the GALEN, GO, and Anatomy ontologies. Instead of splitting the axioms into separate training, validation, and testing sets, we now train the models on the entire ontology using all of its asserted axioms. For evaluation, we use the ELK reasoner [19] to compute the complete set of NF1 axioms (i.e., atomic subsumptions) that are logically implied by, but not explicitly asserted in the given ontology. We split off 10% of these implied NF1 axioms for the validation set and keep the remainder as the testing set. We report the results of Box$^2$EL and the same baseline methods considered in subsumption prediction. The evaluation and experimental protocol is also the same as in subsumption prediction.[5]

---

[5]Deductive reasoning with $\mathcal{EL}^{++}$ embeddings has been previously considered in [25]. However, we find that there is significant leakage (overlap) between their testing and training sets. We therefore do not adopt their reported results, but instead reproduce the results of the baseline models.

*Results.* Table 6 lists the results of approximating deductive reasoning. The baselines perform similarly, with ELBE achieving slightly stronger results than the others on GO and Anatomy. Box$^2$EL outperforms the baselines on almost all metrics across the three ontologies, with significant performance gains especially for Hits@100, median rank, and mean rank. This indicates that Box$^2$EL is able to preserve more of the logical structure than the other embedding methods.

*Deductive vs inductive reasoning.* Comparing the results in Tables 1 and 6, we observe that the embedding models generally perform better in subsumption prediction than in approximating deductive reasoning. To see why, note that the learned embeddings are used to make purely statistical predictions about missing axioms. The soundness of our method guarantees that these predictions align with the semantics of the ontology. However, we do not explicitly perform logical inference steps in the embedding space, as would be required to derive logical inferences similar to a deductive reasoning algorithm. We illustrate this difference with a concrete example in Appendix F. While embedding methods can thus be useful to approximate deductive reasoning, the two approaches are best used in conjunction in order to combine formal derivations with inductive and probable knowledge.

## 5 CONCLUSION AND FUTURE WORK

We developed Box$^2$EL, a novel OWL ontology embedding method that adopts box-based representations for both concepts and roles. This representation is able to model complex logical constructs from $\mathcal{EL}^{++}$ and overcomes the limitations of previous approaches in representing roles and role inclusion axioms. We formally proved that our method is sound, i.e., correctly represents the semantics of $\mathcal{EL}^{++}$, and performed an extensive empirical evaluation, achieving state-of-the-art results in concept subsumption prediction, role assertion prediction, and approximating deductive reasoning. We additionally introduced a new benchmark for evaluating subsumption prediction involving complex concepts.

Our approach offers several interesting directions for future work. For example, we plan to investigate extending Box$^2$EL with additional logical constructs. Furthermore, we also plan to apply our model to other real-world life science applications such as gene-disease association prediction.

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

## A  PROOF OF THEOREM 1 (SOUNDNESS)

In order to prove Theorem 1, we first show the correctness of our loss functions. Recall that the inclusion loss $\mathcal{L}_{\subseteq}(A, B)$ of two boxes $A$ and $B$ is defined as

$$\mathcal{L}_{\subseteq}(A, B) = \begin{cases} \|\max\{\mathbf{0}, \, \mathbf{d}(A, B) + 2\mathbf{o}(A) - \gamma\}\| & \text{if } B \neq \emptyset \\ \max\{0, \mathbf{o}(A)_1 + 1\} & \text{otherwise.} \end{cases}$$

LEMMA 1. *Let $A$ and $B$ be boxes in $\mathbb{R}^n$. If $\gamma \leq 0$ and $\mathcal{L}_{\subseteq}(A, B) = 0$, then $A \subseteq B$.*

PROOF. First, assume that $B = \emptyset$. Since $\mathcal{L}_{\subseteq}(A, B) = 0$, we have $\mathbf{o}(A)_1 = \mathbf{u}_{A,1} - \mathbf{l}_{A,1} \leq -1$ and thus $\mathbf{l}_{A,1} > \mathbf{u}_{A,1}$. Therefore $A = \emptyset$.

If $B \neq \emptyset$, we show that $\mathbf{l}_B \leq \mathbf{l}_A$ and $\mathbf{u}_A \leq \mathbf{u}_B$. Assume $\mathcal{L}_{\subseteq}(A, B) = 0$. We have that

$$\mathbf{d}(A, B) + 2\mathbf{o}(A) - \gamma \leq 0$$
$$|\mathbf{c}(A) - \mathbf{c}(B)| + \mathbf{o}(A) - \mathbf{o}(B) - \gamma \leq 0$$

and thus

$$|\mathbf{c}(A) - \mathbf{c}(B)| + \mathbf{o}(A) - \mathbf{o}(B) \leq \gamma \leq 0.$$

Now fix an arbitrary dimension $k$ such that $1 \leq k \leq n$. We distinguish two cases:

**Case 1:** $\mathbf{c}(A)_k \geq \mathbf{c}(B)_k$. We eliminate the absolute value function and use $\mathbf{u}_\beta = \mathbf{c}(\beta) + \mathbf{o}(\beta)$ for an arbitrary box $\beta$ to obtain

$$\mathbf{u}_{A,k} - \mathbf{u}_{B,k} \leq 0$$
$$\mathbf{u}_{A,k} \leq \mathbf{u}_{B,k}.$$

Since $\mathbf{c}(A)_k \geq \mathbf{c}(B)_k$, we furthermore have by the definition of $\mathbf{c}(\cdot)$ that

$$\frac{\mathbf{l}_{A,k} + \mathbf{u}_{A,k}}{2} \geq \frac{\mathbf{l}_{B,k} + \mathbf{u}_{B,k}}{2}$$
$$\mathbf{l}_{A,k} \geq \mathbf{l}_{B,k} + \underbrace{\mathbf{u}_{B,k} - \mathbf{u}_{A,k}}_{\geq 0}$$
$$\mathbf{l}_{A,k} \geq \mathbf{l}_{B,k}.$$

**Case 2:** $\mathbf{c}(A)_k \leq \mathbf{c}(B)_k$. Similarly to the first case, we eliminate the absolute value function and use $\mathbf{l}_\beta = \mathbf{c}(\beta) - \mathbf{o}(\beta)$ to obtain

$$-\mathbf{l}_{A,k} + \mathbf{l}_{B,k} \leq 0$$
$$\mathbf{l}_{B,k} \leq \mathbf{l}_{A,k}.$$

Because $\mathbf{c}(A)_k \leq \mathbf{c}(B)_k$, we furthermore have

$$\frac{\mathbf{l}_{A,k} + \mathbf{u}_{A,k}}{2} \leq \frac{\mathbf{l}_{B,k} + \mathbf{u}_{B,k}}{2}$$
$$\underbrace{\mathbf{l}_{A,k} - \mathbf{l}_{B,k}}_{\geq 0} + \mathbf{u}_{A,k} \leq \mathbf{u}_{B,k}$$
$$\mathbf{u}_{A,k} \leq \mathbf{u}_{B,k}.$$

Now, consider an arbitrary point $\mathbf{a} \in A$, i.e., $\mathbf{l}_A \leq \mathbf{a} \leq \mathbf{u}_A$. But then

$$\mathbf{l}_B \leq \mathbf{l}_A \leq \mathbf{a} \leq \mathbf{u}_A \leq \mathbf{u}_B$$

and thus $\mathbf{a} \in B$. □

LEMMA 2. *Let $A$ and $B$ be boxes in $\mathbb{R}^n$. If $\gamma \leq 0$ and*

$$\|\max\{\mathbf{0}, -(\mathbf{d}(A, B) + \gamma)\}\| = 0,$$

*then $A \cap B = \emptyset$.*

PROOF. The proof is similar to that of Lemma 1. We have that

$$-(\mathbf{d}(A, B) + \gamma) \leq 0$$
$$-(|\mathbf{c}(A) - \mathbf{c}(B)| - \mathbf{o}(A) - \mathbf{o}(B) + \gamma) \leq 0$$

and therefore

$$|\mathbf{c}(A) - \mathbf{c}(B)| - \mathbf{o}(A) - \mathbf{o}(B) \geq -\gamma \geq 0.$$

We again fix a dimension $k$ such that $1 \leq k \leq n$ and distinguish two cases:

**Case 1:** $\mathbf{c}(A)_k \geq \mathbf{c}(B)_k$. Eliminating the absolute value function yields

$$\mathbf{l}_{A,k} - \mathbf{u}_{B,k} \geq 0$$
$$\mathbf{l}_{A,k} \geq \mathbf{u}_{B,k}.$$

**Case 2:** $\mathbf{c}(A)_k \leq \mathbf{c}(B)_k$. Analogously to Case 1, we have

$$\mathbf{l}_{B,k} - \mathbf{u}_{A,k} \geq 0$$
$$\mathbf{l}_{B,k} \geq \mathbf{u}_{A,k}.$$

Now consider an arbitrary point $\mathbf{a} \in A$. From the case analysis above, we know that either $\mathbf{a}_k \geq \mathbf{u}_{B,k}$ or $\mathbf{a}_k \leq \mathbf{l}_{B,k}$. However, in both cases $\mathbf{a}$ cannot be in $B$. □

We are now ready to prove Theorem 1, restated below.

THEOREM 1 (SOUNDNESS). *Let $O = (\mathcal{T}, \mathcal{A})$ be an $\mathcal{EL}^{++}$ ontology. If $\gamma \leq 0$ and there exist Box$^2$EL embeddings in $\mathbb{R}^n$ such that $\mathcal{L}(O) = 0$, then these embeddings are a model of $O$.*

PROOF. We first perform the standard steps of transforming the ABox and normalizing the axioms in $O$. Let $O'$ denote the resulting ontology. The Box$^2$EL embeddings induce the geometric interpretation $\mathcal{I} = (\Delta^\mathcal{I}, \cdot^\mathcal{I})$ defined as follows:

(1) $\Delta^\mathcal{I} = \mathbb{R}^n$,
(2) for every concept $C \in \mathcal{N}_C$, let $C^\mathcal{I} = \text{Box}(C)$,
(3) for every individual $a \in \mathcal{N}_I$, let $a^\mathcal{I} = \mathbf{e}_a$,
(4) for every role $r \in \mathcal{N}_R$, let $r^\mathcal{I} = \text{Head}(r) \times \text{Tail}(r)$.

We show that $\mathcal{I}$ is a model of $O'$. First, note that $\mathcal{L}(O) = 0$ implies that the regularization loss is 0, and thus $\text{Bump}(C) = \mathbf{0}$ for any $C \in \mathcal{N}_C \cup \mathcal{N}_I$. We now show that $\mathcal{I}$ satisfies every axiom $\alpha \in O'$, distinguishing between the different normal forms. Implicitly, we make frequent use of Lemma 1, which we do not state explicitly for the sake of brevity.

**Case 1:** $\alpha = C \sqsubseteq D$. Since $\mathcal{L}_1(C, D) = 0$ and therefore $\mathcal{L}_{\subseteq}(\text{Box}(C), \text{Box}(D)) = 0$, we have that $\text{Box}(C) \subseteq \text{Box}(D)$. But then it immediately follows from the definition of $\mathcal{I}$ that $C^\mathcal{I} \subseteq D^\mathcal{I}$.

**Case 2:** $\alpha = C \sqcap D \sqsubseteq E$. We have that $\mathcal{L}_2(C, D, E) = 0$ and therefore it follows that $\text{Box}(C) \cap \text{Box}(D) \subseteq \text{Box}(E)$. Hence, we have $(C \sqcap D)^\mathcal{I} = C^\mathcal{I} \cap D^\mathcal{I} = \text{Box}(C) \cap \text{Box}(D) \subseteq \text{Box}(E) = E^\mathcal{I}$.

**Case 3:** $\alpha = C \sqsubseteq \exists r.D$. Assume $D^\mathcal{I} \neq \emptyset$. Let $x \in C^\mathcal{I} = \text{Box}(C)$. Since $\mathcal{L}_3(C, r, D) = 0$ and all bump vectors are $\mathbf{0}$, we have $\text{Box}(C) \subseteq \text{Head}(r)$ and therefore $x \in \text{Head}(r)$. Similarly, for any $y \in D^\mathcal{I}$ we have $y \in \text{Tail}(r)$. But then $(x, y) \in r^\mathcal{I}$ and therefore $x \in (\exists r.D)^\mathcal{I}$.

If on the other hand $D^\mathcal{I} = \emptyset$, we also have $(\exists r.D)^\mathcal{I} = \emptyset$. Since $\text{Box}(D)$ is empty, we furthermore have $\mathcal{L}_{\subseteq}(\text{Box}(C), \emptyset) = 0$ and therefore $\text{Box}(C) = \emptyset$, i.e., $C^\mathcal{I} \subseteq (\exists r.D)^\mathcal{I}$.

**Case 4:** $\alpha = \exists r.C \sqsubseteq D$. Assume $D^\mathcal{I} \neq \emptyset$. Let $x \in (\exists r.C)^\mathcal{I}$. Hence, there exist a $y \in C^\mathcal{I}$ such that $(x, y) \in r^\mathcal{I}$. By the definition of

**Table 7: Sizes of the different ontologies we consider. The number of classes, roles, and axioms in each normal form is reported.**

| Ontology | Classes | Roles | $C \sqsubseteq D$ | $C \sqcap D \sqsubseteq E$ | $C \sqsubseteq \exists r.D$ | $\exists r.C \sqsubseteq D$ | $C \sqcap D \sqsubseteq \bot$ | $r \sqsubseteq s$ | $r_1 \circ r_2 \sqsubseteq s$ |
|---|---|---|---|---|---|---|---|---|---|
| GALEN | 24,353 | 951 | 28,890 | 13,595 | 28,118 | 13,597 | 0 | 958 | 58 |
| GO | 45,907 | 9 | 85,480 | 12,131 | 20,324 | 12,129 | 30 | 3 | 6 |
| Anatomy | 106,495 | 188 | 122,142 | 2,121 | 152,289 | 2,143 | 184 | 89 | 31 |

**Table 8: Impact of the role representation on the performance of Box²EL. We compare our model with a version in which roles are represented as translations (Box²EL-Tr).**

| Model | H@1 | H@10 | H@100 | Med | MRR | MR | AUC |
|---|---|---|---|---|---|---|---|
| Box²EL-Tr | 0.03 | 0.15 | 0.30 | 1141 | 0.07 | 4793 | 0.79 |
| Box²EL | **0.05** | **0.20** | **0.35** | **669** | **0.10** | **4375** | **0.81** |

$r^{\mathcal{I}}$, we must therefore have $x \in \text{Head}(r)$. Since $\mathcal{L}_4(r, C, D) = 0$, furthermore $\text{Head}(r) \subseteq \text{Box}(D)$ and therefore $x \in D^{\mathcal{I}}$.

If on the other hand $D^{\mathcal{I}} = \emptyset$, we have $\text{Head}(r) \subseteq \emptyset$ and thus $r = \emptyset$, so $(\exists r.C)^{\mathcal{I}} = \emptyset$.

**Case 5:** $\alpha = C \sqcap D \sqsubseteq \bot$. We have $\mathcal{L}_5(C, D) = 0$, so by Lemma 2 we have that $(C \sqcap D)^{\mathcal{I}} = \text{Box}(C) \cap \text{Box}(D) = \emptyset \subseteq \bot^{\mathcal{I}}$.

**Case 6:** $\alpha = r \sqsubseteq s$. Let $(a, b) \in r^{\mathcal{I}}$. By the definition of $r^{\mathcal{I}}$, $a \in \text{Head}(r)$ and $b \in \text{Tail}(r)$. Since $\mathcal{L}_6(r, s) = 0$, we furthermore have $\text{Head}(r) \subseteq \text{Head}(s)$ and $\text{Tail}(r) \subseteq \text{Tail}(s)$ and hence $(a, b) \in s^{\mathcal{I}}$.

**Case 7:** $\alpha = r_1 \circ r_2 \sqsubseteq s$. Let $(a, b) \in r_1^{\mathcal{I}} \circ r_2^{\mathcal{I}}$. By definition, we have $a \in \text{Head}(r_1)$ and $b \in \text{Tail}(r_2)$. Because $\mathcal{L}_7(r_1, r_2, s) = 0$, $\text{Head}(r_1) \subseteq \text{Head}(s)$ and $\text{Tail}(r_2) \subseteq \text{Tail}(s)$, so $(a, b) \in s^{\mathcal{I}}$.

We have shown that $\mathcal{I}$ satisfies every axiom in $O'$, and is therefore a model of $O'$. But since $O'$ is a conservative extension of $O$ [3], it follows that $\mathcal{I}$ is also a model of $O$. □

## B  STATISTICAL INFORMATION ABOUT BENCHMARK ONTOLOGIES

The sizes of the ontologies we consider in terms of number of classes, roles, and axioms are reported in Table 7.

## C  ABLATION STUDIES

We conduct two ablation studies to investigate the performance impact of different parts of our model. All studies are conducted on the GALEN ontology for the subsumption prediction task and we report results combined across all normal forms.

### C.1  Impact of Role Representation

We consider an alternative model that uses the exact same optimization procedure as Box²EL, but represents roles as translations, similar to previous methods. The results are listed in Table 8.

We observe that the model that represents roles as translations performs worse on all metrics, in most cases by a large margin. The results highlight the importance of the novel role representation for the performance of our model.

**Table 9: Impact of the number of negative samples on the performance of Box²EL. The model Box²EL-$\omega$ denotes Box²EL trained with $\omega$ negative samples per NF3 axiom.**

| Model | H@1 | H@10 | H@100 | Med | MRR | MR | AUC |
|---|---|---|---|---|---|---|---|
| Box²EL-0 | 0.00 | 0.01 | 0.06 | 7351 | 0.01 | 8727 | 0.62 |
| Box²EL-1 | **0.05** | **0.20** | **0.35** | 676 | **0.10** | 4397 | 0.81 |
| Box²EL-2 | **0.05** | **0.20** | **0.35** | 638 | **0.10** | 4255 | **0.82** |
| Box²EL-3 | **0.05** | 0.19 | **0.35** | **625** | 0.09 | 4187 | **0.82** |
| Box²EL-4 | **0.05** | 0.19 | **0.35** | 628 | **0.10** | 4177 | **0.82** |
| Box²EL-5 | **0.05** | 0.19 | **0.35** | 627 | **0.10** | **4174** | **0.82** |

### C.2  Number of Negative Samples

Our second ablation study concerns the performance impact of the number of negative samples. We report results for Box²EL models trained with 0-5 negative samples per NF3 axiom in Table 9.

We observe that the model that uses no negative samples performs significantly worse than the other models, demonstrating that negative sampling is essential to learn strong embeddings. Using more than one negative sample per NF3 axiom further improves the results, but only marginally.

## D  SCORING FUNCTIONS

Scoring functions are used to compute the likelihood of candidate axioms based on the learned embeddings of their concepts. We define a scoring function $s(\cdot)$ for candidate axioms in all four normal forms NF1−4.

*First and second normal form.* For an NF1 axiom $C \sqsubseteq D$, the score is based simply on the distance between the embeddings of $C$ and $D$, i.e., for Box²EL we have

$$s(C \sqsubseteq D) = -\|\boldsymbol{c}(\text{Box}(C)) - \boldsymbol{c}(\text{Box}(D))\|.$$

The same formulation can be used for the baseline methods. Similarly, for NF2 axioms $C \sqcap D \sqsubseteq E$, the score is defined as the negative distance of the embedding of $E$ to the intersection of $C$ and $D$ in the embedding space.

*Third normal form.* For axioms in the third and fourth normal form the scoring function differs between Box²EL and the baseline methods, because of the different role representation. For Box²EL, the score for a subsumption $C \sqsubseteq \exists r.D$ is naturally defined as

$$s(C \sqsubseteq \exists r.D) = - \|\boldsymbol{c}(\text{Box}(C)) + \text{Bump}(D) - \boldsymbol{c}(\text{Head}(r))\|$$
$$- \|\boldsymbol{c}(\text{Box}(D)) + \text{Bump}(C) - \boldsymbol{c}(\text{Tail}(r))\|.$$

In the baseline methods, the score is computed similarly to TransE:

$$s(C \sqsubseteq \exists r.D) = -\|\boldsymbol{c}(\text{Box}(C)) + \boldsymbol{v}(r) - \boldsymbol{c}(\text{Box}(D))\|,$$

where $\boldsymbol{v}(r)$ is the embedding of role $r$.

*Fourth normal form.* Finally, for an NF4 axiom $\exists r.C \sqsubseteq D$, the score assigned by Box$^2$EL is given by

$$s(\exists r.C \sqsubseteq D) = -\|\boldsymbol{c}(\text{Head}(r)) - \text{Bump}(C) - \boldsymbol{c}(\text{Box}(D))\|,$$

and for the baseline methods we have

$$s(\exists r.C \sqsubseteq D) = -\|\boldsymbol{c}(\text{Box}(C)) - v(r) - \boldsymbol{c}(\text{Box}(D))\|.$$

*Volume-based scoring functions.* An alternative, which is employed by BoxEL [44], is to define scoring functions based on volumes instead of distances. We empirically find distance-based scoring functions to perform much better, which is why we adopt this approach in Box$^2$EL.

## E HYPERPARAMETERS

For all experiments, embeddings are learned with the Adam optimizer [20] for a maximum of 10,000 epochs. All hyperparameters are selected based on validation set performance. The values considered are: dimensionality $n \in \{25, 50, 100, 200\}$, margin $\gamma \in \{0, 0.005, 0.02, 0.05, 0.1, 0.15\}$, learning rate lr $\in \{5 \times 10^{-2}, 1 \times 10^{-2}, 5 \times 10^{-3}, 1 \times 10^{-3}, 5 \times 10^{-4}\}$, negative sampling distance $\delta$ in steps of size 0.5 from 0 to 5.5, number of negative samples $\omega \in \{1, 2, 3, 4, 5\}$, and regularization $\lambda \in \{0, 0.05, 0.1, 0.2, 0.3, 0.4, 0.5\}$.

We report the hyperparameters used in the experiments in Table 10. Note that for ELBE on GALEN in subsumption prediction, and for Box$^2$EL on GALEN in deductive reasoning and Yeast in PPI prediction, we furthermore decay the learning rate with a factor of 0.1 after 2000 training epochs. For PPI prediction, we only report the hyperparameters for Box$^2$EL, since the baseline results are taken from the literature.

## F COMPARISON OF DEDUCTIVE AND INDUCTIVE REASONING

We investigate the comparatively lower performance of the embedding models on deductive reasoning compared to the inductive subsumption prediction task with a concrete example. In the GALEN ontology, we find the following subsumption axiom to predict in the testing set:

$$\text{SodiumLactate} \sqsubseteq \text{NAMEDComplexChemical}.$$

This axiom cannot be logically inferred from the training data. However, the following similar subsumptions are part of the training data:

$$\text{SodiumLactate} \sqsubseteq \exists\text{isMadeOf.Sodium}$$
$$\text{SodiumBicarbonate} \sqsubseteq \exists\text{isMadeOf.Sodium}$$
$$\text{SodiumCitrate} \sqsubseteq \exists\text{isMadeOf.Sodium}$$
$$\text{SodiumBicarbonate} \sqsubseteq \text{NAMEDComplexChemical}$$
$$\text{SodiumCitrate} \sqsubseteq \text{NAMEDComplexChemical}.$$

It seems quite likely that our model will be able to exploit the statistical information contained in these subsumptions to learn embeddings where SodiumLactate is close to NAMEDComplexChemical, and thus achieve a high score for the desired true axiom.

In contrast, in the deductive reasoning setting, all testing axioms are logical inferences of the training data. One such testing axiom is

$$\text{SodiumLactate} \sqsubseteq \text{SodiumCompound}.$$

This logical inference has been reached by a reasoning algorithm such as ELK [19] by performing the sequence of derivations listed in Figure 4. Embedding methods, on the other hand, do not explicitly perform these derivations. Instead, they learn embeddings that align with the semantics of the ontology, i.e., in convergence we will have that Box(SodiumLactate) ⊆ Box(SodiumCompound). However, our model has to learn highly accurate embeddings for a number of different concepts and roles involved in the derivation sequence for this to actually hold in practice. Furthermore, the model then makes statistical predictions based on the scoring functions instead of only predicting axioms that completely align with the semantics. Hence there may be many other plausible axioms that are ranked higher by the model, decreasing the score of the desired logical inference.

Received 20 February 2007; revised 12 March 2009; accepted 5 June 2009

**Table 10: Hyperparameters for subsumption prediction (a), deductive reasoning (b), and PPI prediction (c).**

**(a)**

| | Model | $n$ | $\gamma$ | lr | $\delta$ | $\omega$ | $\lambda$ |
|---|---|---|---|---|---|---|---|
| GALEN | ELEm | 200 | 0.05 | $5 \times 10^{-4}$ | – | – | – |
| | EmEL$^{++}$ | 200 | 0.05 | $5 \times 10^{-4}$ | – | – | – |
| | BoxEL | 50 | – | $1 \times 10^{-3}$ | – | – | – |
| | ELBE | 200 | 0.05 | $5 \times 10^{-3}$ | – | – | – |
| | Box$^2$EL | 200 | 0.15 | $1 \times 10^{-2}$ | 5 | 1 | 0.4 |
| GO | ELEm | 200 | 0.1 | $5 \times 10^{-4}$ | – | – | – |
| | EmEL$^{++}$ | 200 | 0.1 | $5 \times 10^{-4}$ | – | – | – |
| | BoxEL | 25 | – | $1 \times 10^{-2}$ | – | – | – |
| | ELBE | 200 | 0.1 | $5 \times 10^{-3}$ | – | – | – |
| | Box$^2$EL | 200 | 0.15 | $1 \times 10^{-2}$ | 5.5 | 5 | 0.5 |
| Anatomy | ELEm | 200 | 0.05 | $5 \times 10^{-4}$ | – | – | – |
| | EmEL$^{++}$ | 200 | 0.05 | $5 \times 10^{-4}$ | – | – | – |
| | BoxEL | 25 | – | $1 \times 10^{-3}$ | – | – | – |
| | ELBE | 200 | 0.05 | $5 \times 10^{-3}$ | – | – | – |
| | Box$^2$EL | 200 | 0.05 | $1 \times 10^{-2}$ | 5.5 | 3 | 0.3 |

**(b)**

| | Model | $n$ | $\gamma$ | lr | $\delta$ | $\omega$ | $\lambda$ |
|---|---|---|---|---|---|---|---|
| GALEN | ELEm | 200 | 0 | $5 \times 10^{-4}$ | – | – | – |
| | EmEL$^{++}$ | 200 | 0 | $5 \times 10^{-4}$ | – | – | – |
| | BoxEL | 25 | – | $5 \times 10^{-3}$ | – | – | – |
| | ELBE | 200 | 0.15 | $1 \times 10^{-2}$ | – | – | – |
| | Box$^2$EL | 200 | 0.05 | $5 \times 10^{-3}$ | 1 | 2 | 0 |
| GO | ELEm | 200 | 0.1 | $1 \times 10^{-3}$ | – | – | – |
| | EmEL$^{++}$ | 200 | 0.1 | $1 \times 10^{-3}$ | – | – | – |
| | BoxEL | 25 | – | $1 \times 10^{-3}$ | – | – | – |
| | ELBE | 200 | 0.05 | $5 \times 10^{-3}$ | – | – | – |
| | Box$^2$EL | 200 | 0.05 | $5 \times 10^{-3}$ | 3 | 3 | 0.05 |
| Anatomy | ELEm | 200 | 0.05 | $5 \times 10^{-4}$ | – | – | – |
| | EmEL$^{++}$ | 200 | 0.05 | $5 \times 10^{-4}$ | – | – | – |
| | BoxEL | 25 | – | $1 \times 10^{-3}$ | – | – | – |
| | ELBE | 200 | 0.05 | $5 \times 10^{-3}$ | – | – | – |
| | Box$^2$EL | 200 | 0.05 | $1 \times 10^{-3}$ | 2 | 2 | 0.05 |

**(c)**

| | $n$ | $\gamma$ | lr | $\delta$ | $\omega$ | $\lambda$ |
|---|---|---|---|---|---|---|
| Yeast | 200 | 0.02 | $1 \times 10^{-2}$ | 2.5 | 4 | 0.2 |
| Human | 200 | 0.005 | $5 \times 10^{-2}$ | 3.5 | 5 | 0.3 |

$$\frac{\text{SodiumLactate} \sqsubseteq \text{NAMEDComplexChemical} \qquad \text{NAMEDComplexChemical} \sqsubseteq \text{ChemicalSubstance}}{\text{SodiumLactate} \sqsubseteq \text{ChemicalSubstance}} \qquad (2)$$

$$\text{SodiumLactate} \sqsubseteq \exists\text{isMadeOf.Sodium} \qquad (3)$$

$$\frac{(2) \qquad (3) \qquad \text{ChemicalSubstance} \sqcap \exists\text{isMadeOf.Sodium} \sqsubseteq \text{SodiumCompound}}{\text{SodiumLactate} \sqsubseteq \text{SodiumCompound}} \qquad (4)$$

**Figure 4: Inference steps required to derive** SodiumLactate $\sqsubseteq$ SodiumCompound. **The derivations are from top to bottom, similar to natural deduction.**

