# OpenReview forum: "Dual Box Embeddings for the Description Logic EL++"
_ACM.org/TheWebConf/2024/Conference — TheWebConf24 Oral_

### Official Review · Reviewer_hKJH · 2023-11-08

**Novelty:** 6
**Technical Quality:** 6

**Review:**

This paper proposes Box2EL, a novel approach for ontology embeddings, supporting the expressivity of the Description Logic EL++. While previous approaches are based on hyperspheres and have some problems representing one-to-many, many-to-one, and many-to-many relations, the novel approach is instead based on hyperrectangles.

The research problem is important and relevant for the conference audience. The approach is novel, well compared with the related work, and well presented. The technical work is very good: the approach is formally presented, proving the soundness, and evaluated on 3 tasks (general subsumption prediction, link prediction, and deductive reasoning). Results are good, outperforming the baseline competitors, and some significant conclusions are identified.

Although the paper is very good in general, there are some points that could be improved.

1. It is not clear why the authors do not always use the same reasoners. jcel is used to transform ontologies into normal form axioms, but ELK to compute the complete set of atomic subsumptions.

2. In Section 2.1, the authors should mention that the logic EL++ also includes concrete domains, which are not discussed in this paper. Actually, in a more recent paper, the authors of EL++ used the same name to refer to a logic including domain and range role axioms:

	Franz Baader, Sebastian Brandt, Carsten Lutz. Pushing the $\mathcal{EL}$ envelope further. In Proceedings of the 4th Workshop on OWL: Experiences and Directions (OWLED 2008 DC), volume 496 of CEUR Workshop Proceedings, 2008.

3. I think that the paper would improve by using a single running example, replacing Example 1 with the axioms in Figure 2, and Figure 1 with Figure 3.

**Questions:**

Why not using always either jcel or ELK?

**Reviewer Confidence:**

3: The reviewer is confident but not certain that the evaluation is correct

**Scope:**

4: The work is relevant to the Web and to the track, and is of broad interest to the community

---

### Official Review · Reviewer_YHbL · 2023-11-17

**Novelty:** 4
**Technical Quality:** 3

**Review:**

The paper investigates a method based on ontology embeddings in order to identify plausible subsumption relations or relationships between individuals, and to perform approximate deductive reasoning.

The paper considers the EL++ description logic, which provides the formal foundations for the EL profile of the W3C’s OWL2 standard. This is achieved through embeddings of an ontology as a geometric representation of hyperrectangles, to which corresponds a first-order logic model of the ontology.
The learning method is introduced, its training is described and the fact that it corresponds to a first-order logic model of the ontology under some assumptions is stated.
Then, the proposed learning model is experimentally evaluated based on several ontologies: GALEN, Gene Ontology, Anatomy, etc. The reported experiments show that the proposed learning method performs better than state-of-the-art competitors for subsumption prediction (whether a subsumption relation may hold between concepts), link prediction (whether a relation may exist between two individuals), and approximation deductive reasoning.

The paper has some strengths:
1/ The topic of the paper is relevant to the Semantics and Knowledge track of WWW.

2/ The paper addresses an interesting issue: whether ontology embeddings can be used to perform approximate reasoning on ontologies.

However, the paper also suffers from some several weaknesses:
1/ The motivations of the paper are not clear.
Although it is interesting to study whether ontology embeddings can be used to perform approximate reasoning on ontologies, motivating the study by the claim that standard logical reasoning is “often too rigid for real-world OWL ontologies and cannot derive knowledge that is only probable from the given data” does not make sense in my opinion. There is a huge literature in IA, notably in knowledge representation and reasoning as well as in symbolic machine learning, that uses logic-based reasoning procedure to infer probable knowledge from the data. I therefore suggest to find more convincing motivations and arguments. In my view, studying whether ontology embeddings can be used to perform approximate reasoning on ontologies is interesting enough per se.

2/ The paper is not self-contained.
Key information is missing to understand the paper and assess its merit.
First, the EL++ description logic used in the paper is neither clearly nor formally defined. Some constructs are syntactically discussed, while other appear without being discussed at all (e.g., role composition), and the semantics of EL++ is not discussed at all. This is problematic as Theorem 1 claims that the proposed embeddings correspond to a first-order model of an EL++ ontology, while this notion has not been properly defined. Currently, the semantics of the EL++ constructors and axioms is missing. We are just told “See Baader et al. [3] for a formal definition”.
Second, the description of the proposed method is not sufficiently detailed. While the description of the geometric representations of concepts and of individuals is ok, the description of the geometric representation of roles uses the central notion of bump vectors that are not defined, as well as of an operation that translates a box along a bump vector which is also undefined. Then, the training procedure relies on a normalization procedure that is not discussed at all. We are referred to [3] for the used procedure. It is important to state some properties of the normalization: the size of a normalized ontology and how the normalized ontology semantically relates to the original one wrt first-order logic.
Finally, the proof of the main technical result is not self-contained: we follow the construction of a model of a normalized ontology from the embeddings. But the last step that must show that this model is or can lead to a model of the original ontology, is reduced to a single phrase that refers to the undefined notion conservative extension for which we are referred to [3].

3/ The paper is rather incremental wrt [22]. The proposed method builds on [22] and follows the same logic of presentation (organization, running example about the same family domain, similar theorem for soundness of the method, and similar evaluation setting). The notable difference is the use of boxes (hyperrectangles) instead of balls in the representation of the learning model, and of (undefined) bumping boxes and vectors for translation. The novelty and originality of the work is thus quite limited in my opinion.

4/ The paper claims to introduce a novel benchmark for subsumption prediction. I do not understand this claim. The benchmark only consists in splitting the known GALEN, Gene Ontology and Anatomy ontologies into training, validation and testing sets of axioms. This does not make a novel benchmark in my opinion. Did I miss something?

Minor remark:
In Section 2.2, the central problem of concept subsumption in DLs is between two complex concepts. The concept subsumption problems discussed in 2.2, between two atomic concepts or between an atomic concept and a complex concept, are particular cases.

**Questions:**

Can you provide formal definitions for bumping boxes/vectors and on the \oplus operation that translates a box along a bump vector?

Can you provide the EL++ normalization that is used and its properties wrt output size and first-order semantics?

Can you provide a self-contained end of the proof of Theorem 1?

Does the novelty of your benchmark go beyond splitting well-known ontologies into training, validation, and testing sets? If yes, can you provide details, please?

**Ethics Review Description:**

No issue

**Reviewer Confidence:**

3: The reviewer is confident but not certain that the evaluation is correct

**Scope:**

4: The work is relevant to the Web and to the track, and is of broad interest to the community

---

### Official Review · Reviewer_TkiM · 2023-11-23

**Novelty:** 5
**Technical Quality:** 5

**Review:**

The paper introduces an ontology embedding method (Box^2EL) for completion of EL++ ontologies, based on the notion of relations and concepts as "boxes" and a bumping mechanism, with the aim of overcoming the current limitations on reasoning on complex role relationships.

The paper first establishes the background on EL++ and the reasoning problems, briefly summarizing the related approaches. The Box^2EL method is then presented, by first defining the geometric model and the training method and then showing the soundess of the method. The paper concludes with an evaluation of the method over general subsumption prediction, link prediction and approximation of deductive reasoning over known EL ontologies.

In general, the positive aspects of the paper are:
- The method has been shown to be sound with the semantic definition of the underlying EL++ description logic.
- Extensive evaluation over different reasoning tasks and existing EL ontologies.
- Code and data will be made available for reproducibility.

On the other hand, some of the negative aspects of the paper are:
- Some of the formal aspects of the method should be better detailed: for example, it is not clear where the normal form for EL++ is defined (and the method to transform any EL++ ontology in this form).
- In the example ontology in section 4.1, the more advanced constructs of EL++ (complex role inclusions) are not used.
- The discussion and comparison with the related approaches is limited.

The topic is clearly of interest for the track. With respect to the technical quality of the work, the method appears to be sound and in general well explained and motivated. The presentation and writing of the paper is clear and the paper is well-organized. While applying known methods, the contributions of the work are sufficiently novel.

**Questions:**

- The authors show the soundness of the method with respect to the DL semantics: what can be said about the completeness of the method?
- Do the presence/absence of an ABox influences the applicability of the method?

**Ethics Review Description:**

None.

**Reviewer Confidence:**

2: The reviewer is willing to defend the evaluation, but it is likely that the reviewer did not understand parts of the paper

**Scope:**

3: The work is somewhat relevant to the Web and to the track, and is of narrow interest to a sub-community

---

### Official Review · Reviewer_Fid4 · 2023-11-24

**Novelty:** 4
**Technical Quality:** 4

**Review:**

The paper proposes a new ontology embedding for EL++ ontologies that accounts for role inclusion and concept intersection.
The work is very relevant, thoroughly evaluated and the results obtained are promising. The presentation, however, needs to be more accessible, not everything is self-contained and easily followed by others like me who are interested in ontologies and reasoning but may not be familiar with all the details of embeddings.

Strengths:
- relevant and somewhat novel approach
- good evaluation

Weaknesses:
- some of the writing needs improvement to make the paper more accessible
- limitations are not discussed
- the paper puts an emphasis on the soundness of their approach, but the rather lengthy proof is buried in the appendix, while the proof sketch is too high-level. Some expansion of the proof sketch would be helpful.

I did not have a chance to review the soundness proof in the appendix in detail.

Related work: You say that KG embeddings are restricted to predicting relational assertions. But if we're talking about predicting missing triples (which is what I think KG embeddings do), this would include concept inclusion, role inclusion, concept instantiation, etc. because they are just another kinds of links in the KG. So saying that KG embeddings are modeling only role assertion as you do in 2.3 doesn't seem right. Am I missing anything here? Please clarify.

Suggestions/Questions for improving the paper's presentation:
I don't quite understand the bump: in Formula (1) you show its use but it seems to imply that every concept has a single bump. Isn't that problematic? Couldn't the bump be a different one with respect to two different related concepts? E.g. assume there are two role inclusion axioms:
C1 is a subset of Exists r1.D and
C2 is a subset of Exists r2.D
Are both using the same Bump(D) ?
Please clarify.

3.2: the transformation seems to introduce new concepts {a}, {b}, etc. For ontologies with large ABoxes, how scalable is that? Is every individual actually treated as a concept then?

403: what's the margin hyper-parameter? It appears again in your soundness theorem, without even saying what is.

4.2: 632-635: can you at least summarize these ontologies used for evaluation in terms of number of concepts, roles, subsumption axioms, role inclusion axioms, use of the more complex axioms to give the reader a better idea how appropriate they are for your evaluation here.

683-687: please explain what these metrics capture. Without that the paper is not very accessible to many readers.

Table 5: what are raw vs filtered metrics?

I acknowledge the authors' response and believe that this paper would benefit from major revisions as indicated by the amount of questions  and lengthy answers needed to clarify many aspects. I do not think this paper is yet ready for publication.

**Questions:**

See my question in the main review:
1. Are KGs really restricted to role assertions only?
2. Please explain the bump

**Reviewer Confidence:**

2: The reviewer is willing to defend the evaluation, but it is likely that the reviewer did not understand parts of the paper

**Scope:**

3: The work is somewhat relevant to the Web and to the track, and is of narrow interest to a sub-community

---

### Official Review · Reviewer_1r2C · 2023-11-25

**Novelty:** 4
**Technical Quality:** 7

**Review:**

Box^2EL is an ontology embedding approach for the description logic EL++. Concepts and properties are represented as boxes. Every role is associated with a head box and a tail box. Evaluation is performed on GALEN, GO, and Anatomy ontologies for the reasoning tasks such as concept subsumption prediction, role assertion prediction, and approximating deductive reasoning. The results indicate that the proposed approach works well and is able to improve on the existing ontology embedding methods.

Strengths
1) The box representation captures the concept intersection and the one-to-many and many-to-many relations.
2) The writing, in general, is easy to follow, especially with the usage of examples.
3) The literature review is good, with clearly identified gaps and the Box^2EL work is appropriately placed with respect to the current state-of-the-art.

Weaknesses
1) This work does not seem relevant to the Web. The CfP states that the relevance to the Web should be discussed on the first page, but I didn't find such a discussion.
2) What is the difference in approach between Box^2EL and BoxEL [1], [44]? It seems like the concept representation is borrowed from BoxEL and the role representation from [44].
3) The Hits@1 and Hits@10 don't show a major improvement, and in some cases, the existing approaches perform better.

**Questions:**

1) What is the relevance of this work to the Web?
2) A few key phrases were used but not explained - bump vector, offset of a box, visualization loss term. What are they?
3) What is the novel aspect of the benchmark? Is it just splitting the axioms into train, validation and test sets for each normal form?
4) It is mentioned that the link prediction results for the baselines have been obtained from their papers instead of reproducing them. Are you sure that all the baselines ELEm, EmEL++, BoxEL, and ELBE do link prediction? It seems EmEL++ does not do that.
5) I am not sure if simple KG link prediction (the ones used for evaluating KG embeddings) is the same as {a} \sqsubseteq \exists r.{b}. In KG link prediction, the quantifier does not exist. So, for example, in your case, does it matter whether it is a universal or an existential quantifier that is used? Does it matter if we instead have some cardinality associated with the relation?
6) Despite the approach being more faithful to the underlying description logic, there isn't a significant improvement in Hits@1 and Hits@10 (these two are probably the two most practical metrics if used in a real-world setting). What are your comments?

**Reviewer Confidence:**

4: The reviewer is certain that the evaluation is correct and very familiar with the relevant literature

**Scope:**

2: The connection to the Web is incidental, e.g., use of Web data or API

---

### Decision · Program_Chairs · 2024-01-22

**Decision:**

Accept (Oral)

**Comment:**

* The paper seems to be perceived by reviewers as adequately for this conference
* The paper has above average novelty,
* The paper has above average scores in TQ, with only minor clarifications needed for an accepted paper.
* Authors were responsive and reviewers were interactive